# Phage Display-Derived Peptides and Antibodies for Bacterial Infectious Diseases Therapy and Diagnosis

**DOI:** 10.3390/molecules28062621

**Published:** 2023-03-14

**Authors:** Hui Zhao, Dan Nie, Yue Hu, Zhou Chen, Zheng Hou, Mingkai Li, Xiaoyan Xue

**Affiliations:** Department of Pharmacology, Air Force Medical University, Xi’an 710032, China

**Keywords:** phage display, antibacterial peptide, multidrug-resistant, antibodies, virulence factor, library construction, anti-virulence strategy, antibacterial adjuvant

## Abstract

The emergence of antibiotic-resistant-bacteria is a serious public health threat, which prompts us to speed up the discovery of novel antibacterial agents. Phage display technology has great potential to screen peptides or antibodies with high binding capacities for a wide range of targets. This property is significant in the rapid search for new antibacterial agents for the control of bacterial resistance. In this paper, we not only summarized the recent progress of phage display for the discovery of novel therapeutic agents, identification of action sites of bacterial target proteins, and rapid detection of different pathogens, but also discussed several problems of this technology that must be solved. Breakthrough in these problems may further promote the development and application of phage display technology in the biomedical field in the future.

## 1. Introduction

The continuous emergence of multidrug-resistant (MDR) and super-bacteria has brought challenges to antibacterial treatment worldwide [1]. Currently, existing clinical antibiotics cannot fully meet the needs of clinical infection control; however, the development of antibiotics lags behind because the discovery of traditional antibiotics depends on the structural modification of natural products synthesized by bacteria [2], which requires a long time and high investment; in addition, obtaining new antibiotics with novel structure and action mechanism is difficult. The rapid development of genome sequencing technology provides a huge space for the discovery of antibiotics based on new targets [3]. Based on computer analysis, an increasing number of new potential antibacterial targets with unknown or known functions, such as flavin mononucleotide riboswitches and the lysine riboswitch, have been reported [4]. Nevertheless, finding inhibitors of these protein targets, especially inhibitors of unknown functional proteins, presents a challenge.

In 1985, George P. Smith successfully displayed the recombinant peptide at the N-terminal of pIII capsid protein of M13 filamentous bacteriophage, and his discovery laid the foundation for drug discovery based on phage display [5]. Gregory Winter used phage display to obtain antibodies and developed adalimumab, which was the first phage display-derived human monoclonal antibody (mAb) approved by the United States Food and Drug Administration (FDA) in 2002, to treat diseases, such as rheumatoid arthritis, psoriasis, and inflammatory bowel diseases. Smith and Winter won the 2018 Nobel Prize in chemistry for their pioneering work [6]. Phage display selection has been developed into a stable, easy-to-implement, and low-cost method, and is now widely used in immunology, cancer research, drug discovery, epitope mapping, protein–protein interaction, and rapid diagnosis and treatment of infectious diseases [7]. Numerous projects, such as exotoxin-neutralizing antibodies raxibacumab, obiltoxaximab, and bezlotixumab, have entered the clinical or clinical trial stage and have been approved by the FDA for treatment of bacterial infections [8]. This paper mainly reviewed the progress of phage display technology in the antibacterial field, including the discovery of antibacterial agents and the study of possible binding sites of antibacterial target proteins and new bacterial diagnosis technologies.

## 2. Overview of Phage Display Technology

Phage display technology has revolutionized several biological fields because of the use of its relatively fast evolution as a powerful mechanism that enables researchers to quickly identify and isolate antibodies or peptides with high affinity and specificity for targets of interest from a large combinatorial library containing billions of antibodies or peptide fragments [6]. Most protein, peptide, nucleic acid, and carbohydrate targets can be used for phage display. In addition to purified targets, whole cells can be screened [9]. Regardless of the target, the basic screening process of phage display experiment, usually called “biopanning,” is similar and mainly includes “binding—washing—eluting—amplifying”. In general, the biopanning step should be repeated three or four times to enrich phages with target peptides or antibodies [10] (Figure 1A); then, enzyme-linked immunosorbent assay (ELISA) or immunocytochemistry is used to confirm the specific binding of peptides or antibodies to the target.

In addition to screening, library construction is a core foundation of phage display technology. The size and quality of libraries are critical for the final screening results of phage display [11]. Currently, linear random phage peptide libraries (such as New England Biolabs peptide library Ph.D.-7 and Ph.D.-12) are among the most commonly used peptide phage display constructs. In general, the length of foreign proteins ranges from 6 amino acids to 43 amino acids because extremely large peptides can potentially interfere with the infectious activity of phage virus or capsid assembly [12]. The conventional construction method for random phage peptide libraries involves the use of (NNK)n codon degeneracy, where N indicates an equimolar mixture of all four nucleotides (A, G, C, and T), and K denotes a 1:1 mixture of G and T. Based on this method, cyclic peptides, bicyclic peptides, and D-peptide libraries were developed and constructed, and they have been successfully used to identify novel targets that cannot be identified by linear random peptide libraries.

Antibody phage display allows researchers to obtain human antibodies more quickly, because, theoretically, antibodies can be found through in vitro screening without relying on immunity and related restrictions, thereby avoiding the human anti-mouse antibody reaction caused by hybridoma technology [13]. Usually, large combinatorial libraries consisting of variable heavy- and light-chain antibody libraries are constructed and expressed as fusions with one of the phage surface coat proteins for display, whereas the genes for antibodies are contained within phage particles. This strategy allows the isolation of suitable antibodies against the desired antigen from a single library in prokaryotic systems. For the construction of an antibody phage display library, single-chain variable fragments (scFvs) or antigen-binding fragments (Fabs) are generally used [13]; however, Fabs have comparably higher structural stability and can be converted into full-length IgG without impairing their functionality. To produce larger antibody libraries, scientists are attempting to use various new methods for library construction. Human V_H_ domains, immunoglobulins of sharks, and variable domains of camel heavy chains (V_H_Hs) have also gained popularity [14].

## 3. Application of Phage Display in the Antibacterial Field

Compared with the traditional technology for antibacterial drug discovery, the unique advantages of phage display have attracted increasing attention (Figure 1B). (1) Potential antibacterial candidate molecules with high affinity and specificity for any target of interest can be obtained. (2) Phage display can also be used for protein–ligand interaction study to obtain information on target-protein binding sites and their function. (3) Numerous novel screening systems of phage display have been designed for rapid detection and diagnosis of bacteria, which are important for the effective control of MDR bacterial infection.

The targets commonly used for screening can be divided into two categories: specific molecular targets and whole bacterial cells. Molecular targets mainly include virulence factors, drug resistance-related proteins, and molecules necessary for bacterial survival. In comparison with specific molecular targets, whole-cell-based screening may be more complex; however, this type of screening has a potential advantage to recognize cell surface structures that have not been previously identified, and all targets on the cell surface are screened simultaneously in their native physiological context. Next, we will focus on the latest progress of phage display technology in antibacterial drug research, protein interaction, and bacterial diagnosis (Table 1 and Table 2).

### 3.1. Therapeutic Peptides or Antibodies Targeting a Single Molecule

#### 3.1.1. Targeting of Bacterial Virulence Factors

Unlike traditional bacteriostatic or bactericides, anti-virulence agents generally target the key molecules involved in bacterial pathogenic mechanisms and decrease the bacterial pathogenicity, allowing bacteria to be cleared easily by the host immune system; therefore, anti-virulence molecules have minimal significant influence on the survival/adaptability of bacteria and reduce the pressure of inducing bacterial resistance. In addition, they cause no damage nor alteration in the composition of natural host microbiota, which implies improvement in long-term safety. Thus far, a large number of experimental data have confirmed that anti-virulence strategies can be successfully used to combat bacterial infections.

(1)Outer membrane protein U (OmpU)

*Vibrio parahaemolyticus* is a Gram-negative bacteria (G^−^ bacteria), flagellate, curved, rod-shaped, halophilic bacterium that causes three distinct types of diseases: gastroenteritis, wound infection, and septicemia [44]. OmpU is one of the important virulence factors of *V. parahaemolyticus* and present in all *Vibrio* species. In addition to its porin function, OmpU plays crucial roles in modulation of host cellular responses and bacterial pathogenesis [45]. Junfang Yu et al. screened for single-domain antibody fragment (sdAb) candidates that bind to *V. parahaemolyticus* OmpU by using a sdAb phage display library and isolated several positive phage clones, among which clone UAb28 showed high enrichment and affinity and can recognize not only OmpU but also OMPs and *V. parahaemolyticus* whole cells [32]. The results of this study indicate that UAb28 may be an ideal candidate to block the interaction between *V. parahaemolyticus* OmpU and host cells.

(2)The type III secretion system (T3SS)

T3SS is a major virulence factor in *Pseudomonas* and associated with bacterial virulence by injecting effector proteins into the cytosol of host cells [46]. In *Pseudomonas aeruginosa*, T3SS is regulated by SpuE, and knockout of SpuE alone attenuates the cytotoxicity of *P. aeruginosa* by downregulating the expression of most T3SS genes. Yang Zhang et al. used naïve phage-displayed library of human scFvs (huscFvs) and observed that scFv5 can recognize spermidine-bound SpuE with a high apparent affinity (half maximal effective concentration = 1.6 nM for direct binding ELISA); then, they obtained the fusion protein scFv5-MPP by connecting scFv5 with a membrane-penetrating peptide (MPP) (KRKKRKKRK). ScFv5-MPP can inhibit T3SS-mediated cytotoxicity in A549 cells and may be a promising lead candidate for the treatment of *P. aeruginosa* infection [34].

(3)*P. aeruginosa* exotoxin A (ETA)

ETA is one of the most potent bacterial virulence factors produced by *P. aeruginosa*. It is an NAD^+^-diphthamide ADP-ribosyl transferase and can induce the termination of protein synthesis and eventually cause cell death. ETA is a single polypeptide chain of 613 amino acids and consists of three different domains: ETA domain-1 (including domains 1A and 1B), ETA domain-2, and ETA domain-3 [47]. A very effective strategy to block the function of ETA is through the use of antibodies to neutralize it. Kasem Kulkeaw et al. constructed an M13 phage library containing a huscfv gene insert and displayed huscfvs on the phage surface [48]. Sirijan Santajit et al. used recombinant ETA-1A, ETA-3, and full-length ETA and biotinylated peptides at the catalytic site of ETA (ADAITGPEEEGGRLETILGW; P) as antigens, screened phages from the HuscFv phage display library, and obtained three ETA-bound Huscfvs (Huscfv-C41, Huscfv-E44, and Huscfv-P32), which can affect ETA-induced apoptosis in mammalian cells [35]. These Huscfvs can be used either alone or in combination with Huscfv cognates that target other virulence factors of *P. aeruginosa* as a novel strategy for fighting against ETA-mediated disease or an alternative treatment for difficult-to-treat infections.

(4)Urease

*Helicobacter pylori*, a -G^−^ spiral bacterium, is classified as a group 1 carcinogen because its infection initiates several inflammatory states in the gastrointestinal tract, which leads to mucosa-associated lymphoma and gastric cancer. Urease is a crucial factor in *H. pylori*, and it helps the bacterium to tolerate and pass through the harsh acidic environment of the stomach and colonize beneath the mucous lining [49]. Mehdi Fouladi et al. isolated and characterized Ab fragments against the flap region of urease by means of a human semi-synthetic phage antibody library, Tomlinson human library I. They obtained the V_L_ antibody C3, which can bind to the urease large subunit with a KD value of 9.78 × 10^−8^ M and inhibit the activity of urease; thus, it was proposed as a therapeutic agent for the clinical and/or preclinical applications [36].

(5)MrkA

MrkA is a major protein of the type III fimbriae complex of *K. pneumoniae* and associated with host cell attachment and biofilm formation [50]. Qun Wang et al. used wild-type strains Kp1901 and Kp1899 (ATCC) as targets to screen naive huscFv phage-display libraries and acquired two scFv-Fcs (KP3 and KP16). Given that human IgG1 is likely the final drug format, researchers converted scFv-Fcs into IgG1 after the initial screenings. KP3 and KP16 had strong binding capability to Kp29011 through the target MrkA protein and displayed potent opsonophagocytic killing activity against multiple *K. pneumoniae* serotypes. KP3 can prevent *K. pneumoniae* association with lung epithelial cells and reduce the attachment of *K. pneumoniae* to human pulmonary epithelial cells A549; it can also reduce significantly lung burden in mice infected with Kp29011 (O1:K2) and Kp9178 at 15 mg∙kg^−1^ [42].

#### 3.1.2. Targeting Resistance-Associated Proteins

Through inhibition of drug-resistant proteins or enzymes, targeting resistance-associated proteins is an effective way to overcome acquired resistance mechanisms of bacteria and prolong the lifespan of existing antibiotics. These anti-resistance agents, such as clavulanic acid and tazobactam, lack or have very weak antibacterial activity and generally cannot be used alone. However, not all resistance-related molecules are suitable targets for antimicrobial drug development because bacteria are extremely cunning and develop diverse strategies to adapt to environmental changes; therefore, numerous bacteria exhibit MDR mechanisms, and obtaining the desired effect through intervention of a single mechanism may be difficult. Thus far, studies are mainly focused on β-lactamase, which is the main reason for the resistance of bacteria to β-lactam antibiotics. Qiongjing Zou et al. used phage-display random peptide library Ph.D.-12 to find peptides bound to immobilized penicillinase from *Bacillus cereus* [24]. The results showed that polypeptide P2 (NIYTTPWGSNWS) is an effective penicillinase inhibitor that functions in a dose-dependent manner. As the concentration of P2 increased from 0 µM to 200 µM, the apparent maximum reaction rate of penicillinase decreased from 2.24 µM∙s^−1^ to 0.49 µM∙s^−1^. The inhibition behavior of P2 is considered a mixed pattern, that is, P2 can bind to penicillinase and penicillinase–penicillin G complex (Ki = 9.22 µM; Ki’ = 33.12 µM; Ki refers to the binding capability of a competitive inhibitor to the enzyme; Ki’ refers to the binding capability of an uncompetitive inhibitor to the enzyme-substrate complex). P2 may be a useful starting point for the design of novel small-molecule inhibitors for penicillinase.

Verona integron-mediated metallo-β-lactamase (VIM) is a kind of acquired metallo-β-lactamase (MβL). Encoded by Verona integron, VIM-4 can hydrolyze a variety of β-lactam antibiotics, such as carbapenems, penicillins, and cephalosporins. Jean S. Sohier et al. used phage display technique to screen sdAb fragments (V_H_Hs), also known as nanobodies, which can recognize epitopes that are inaccessible to conventional antibodies [14]. After immunizing a dromedary with VIM-4 six times, they obtained a single-cell nanobody NbVIM_38 that can inhibit VIM-4 at the micromole level. They also observed that NbVIM_38 interacted with an epitope that is distant from the active site of VIM-4, altered the VIM-4 substrate binding and catalytic properties, and showed a partial mixed inhibition, which indicated that NbVIM_38 nanobody may be an allosteric inhibitor of VIM-4.

#### 3.1.3. Targeting Proteins Necessary for Bacterial Survival

Different from anti-virulence and anti-resistance strategy, targeting key molecules necessary for bacterial survival has been the mainstream direction of antimicrobial research because this strategy can directly inhibit or kill bacteria and has significant therapeutic value in clinical practice. However, based on traditional antibacterial targets, such as penicillin-binding proteins, finding new compounds that are better than existing antibiotics has been difficult; therefore, developing antibacterial drugs with new antibacterial mechanisms, that is, by acting on new targets, is one of the most important strategies for controlling MDR bacteria.

(1)UDP-N-acetylglucosamine (UDP-GlcNAc) acyltransferase (LpxA)

Lipid A is a hydrophobic anchor that secures the sugar components (core and O-groups) of lipopolysaccharides to the external surface of the outer membrane. LpxA catalyzes the first step of lipid A biosynthesis and the transfer of an R-3-hydroxymyristoyl chain from its acyl carrier protein (ACP) to the 3-OH glucosamine group of UDP-GlcNAc [51]. In 2003, R. Edward Benson et al. discovered a peptide with high affinity toward LpxA (peptide 920) using a recombinant M13 phage display peptide library and verified that the peptide 920-GST fusion protein can significantly inhibit the growth of *Escherichia coli* [18]. In 2006, Allison H. Williams et al. determined the inhibitory effect of peptide 920 on LpxA [19]. The results showed that the half maximal inhibitory concentration (IC_50_) reached 60 ± 9 nM when 1 µM UDP-GlcNAc and 1 µM R-3-hydroxymyristoyl-ACP were used and supported the view that the acyl phosphate alanine portion of peptide 920 and R-3-hydroxyinositol-ACP may compete for the same or overlapping sites on LpxA. Allison H. Williams et al. used peptide 920 as a scaffold and created a small peptide library [20]. Peptide CR20 was obtained by removing three residues at the N-terminal of peptide 920, and it can inhibit LpxA effectively at the IC_50_ of 50 nM. The crystal structure of the complex of *E. coli* LpxA and peptide CR20 was further studied, providing helpful insights into the rational design of LpxA inhibitors.

(2)Histidine-containing phosphorylated carrier protein (HPr)

The phosphotransferase system (PTS) widely exists in fungi and bacteria but is absent in other eukaryotes; it catalyzes the uptake and phosphorylation of related carbohydrates [52]. In addition to sugar transport regulation, the PTS system is involved in chemotaxis, catabolism inhibition, signal transduction, and allosteric regulation of metabolic enzymes and transporters in response to carbohydrate availability [53]. HPr is a small phosphocarrier protein that is essential for the transport and regulatory functions of PTS [54]. Kuo Chih Lin et al. used recombinant HPr as the target and obtained five peptides by screening the phage displayed random peptide library Ph.D.-12 [26]. One of these peptides was AP1 (YQVTQSKVMSHR), which can effectively inhibit the growth of *E. coli* cells at the IC_50_ of 50 µM. AP1 interacts specifically with dephosphorylated HPr (in the presence of high medium glucose concentration) and blocks the interaction between dephosphorylated HPr and glycogen phosphorylase, attenuating the binding capability of HPr to glycogen phosphorylase and preventing the activation of glycogen phosphorylase. This process will slow down the decomposition and accumulation of glycogen in cells and eventually induce cell aggregation; thus, AP1 can be used as a leading peptide for further optimization.

(3)Lipid II and lipid I

The bacterial cell wall is mainly composed of peptidoglycans (PGs), which are three-dimensional networks composed of long amino sugar chains. Lipid II is a membrane-anchored cell wall precursor, and lipid I is the precursor of lipid II, both are essential to the biosynthesis of the bacterial cell wall [55]. Several lipid II inhibitors have been developed and used in clinical settings. Vancomycin can act on the two D-alanines at the end of the pentapeptide side chain of lipid II to prevent the further conversion of lipid II to PG. However, bacteria can reduce the affinity of vancomycin by changing the amino acid at the end of the pentapeptide side chain, which results in clinical drug resistance; therefore, development of antibacterial agents acting on other non-mutable sites of lipid II or I has become a new hotspot in this field.

Heinis and Winter developed a bicyclic variant of the phage display technique by cyclizing random peptides containing three fixed cysteine residues displayed on phage with the chemical reagent 1,3,5-tris(bromomethyl)benzene (TBMB) [56]. Compared with monocyclic peptides of comparable molecular mass, bicyclic peptides bind to their targets with a higher affinity and are more resistant to proteolytic degradation [57,58] because they are more constrained in their conformation. Given that lipids I and II have common key structural elements, and numerous lipid II targeting antibiotics bind lipids I and II with similar affinities, Peter ’t Hart et al. used lipid I and its enantiomers as target molecules to screen a phage bicyclic variant library and obtained peptides with antibacterial activities, such as P8 with a minimum inhibitory concentration (MIC) of 128 µg∙mL^−1^ [22]; then, they used P8 as a lead compound and synthesized a series of P8 variants with C-terminal lipidation. The results showed that P8-D-C10 lipopeptides have strong antibacterial activities against various Gram-positive bacteria (G^+^ bacteria), including a group of vancomycin-resistant strains, such as the vancomycin-resistant *Enterococcus faecium* strain (VRE155) (MIC: 4 µg∙mL^−1^ to 64 µg∙mL^−1^ and little-to-no hemolytic activity). Further studies confirmed that their inhibitory effects on the synthesis of the bacterial cell wall were mediated by targeting and binding with lipid II.

Emel Adaligil et al. used mirror phage display (a process of selecting “D-peptide” with phage display technology [59]) to discover peptide antibiotics composed of D-amino acids [17]. D-Peptides are expected to become more metabolically stable than L-peptides and refractory to protease action. The researchers used five different phage display “D-peptide” libraries and linear peptide libraries Ph.D.-12 and Ph.D.-7; cyclic phage displayed Ph.D.-C7C and bicyclic peptide libraries BC-A and BC-B, which were reconstructed from linear peptide libraries that reacted with TBMB. The targets used in this paper were enantiomers of the pentapeptide precursor of the bacterial cell wall of *Staphylococcus aureus* and enantiomers of derivatives of cephalosporin, which mimics a high-energy conformation of the D-Ala-D-Ala terminus of PG structures in bacterial cell walls. P14 obtained from BC-A showed a strong antibacterial activity against *S. aureus*, methicillin-resistant *S. aureus*, and *Enterococci* (MIC values were 8, 32, and 32 µg∙mL^−1^, respectively). P15 obtained from BC-B exhibited a inhibitory activity against vancomycin-sensitive bacteria (MIC values ranged from 8 µg∙mL^−1^ to 32 µg∙mL^−1^) but was less active against vancomycin-resistant strains. P18 obtained from BC-A exhibited an excellent activity against vancomycin-sensitive bacteria and moderate activity against the vancomycin-resistant *Enterococci* strain (*van*B). P14, P15, and P18 neither caused the lysis of erythrocytes nor showed significant toxicity against mammalian cells (HeLa) at concentrations up to 256 µg∙mL^−1^. The identified peptide antibiotics composed of D-amino acids are orally available and highly stable in human sera and in the presence of protease cocktail trypsin [60]. These data provide a basis for a novel platform for the development of selective, potent, and stable antimicrobial peptides.

(4)Isocitrate lyase (ICL)

Despite the existence of effective combination chemotherapy and neonatal anti-tuberculosis (TB) vaccine, the clinical diagnosis and treatment of TB caused by *Mycobacterium tuberculosis* (MTB) infection are still complex and difficult, which is one of the serious challenges facing global public health [61]. ICL involves in the mycobacterial glyoxylate and methylisocitrate cycles so is an important enzyme for the growth and survival of MTB during latent infection [62]. It is postulated that the inhibition of ICL can disrupt the life cycle of MTB. Theam Soon Lim et al. panned against MTB ICL proteins using a previously developed naive human scFv antibody phage library and identified a single scFv clone (a-rICL-C6) against MTB ICL [30,63]. Further study showed that the a-rICL-C6 monoclonal antibody was able to relieve recombinant ICL enzyme activity with an estimated IC50 of 10 mg∙mL^−1^. In silico prediction and analysis indicated that the inhibition of recombinant ICL by a-rICL-C6 scFv might be due to the disrupted tertiary structure or the formation of the active site β domain. Further validation is needed to confirm if the isolated clone is a good inhibitor against ICL and a new therapeutic agent against MTB.

### 3.2. Targeting Whole Bacterial Cells

Although whole-cell-targeted screening has numerous advantages, the complexity of bacterial cell surface and diversity of expressed molecular species may interfere with the screening of target molecules; therefore, to increase the specificity of screened products, most researchers perform whole-cell subtractive screening, in which two different cells (one target cell and one negative control cell) are used.

#### 3.2.1. *E. coli*

Shilpakala Sainath Rao et al. used phage display technology to screen peptides that bind to the surface of *E. coli* cells [25]. They first used a subtractive phage-display approach where the phage-display random-peptide library Ph.D.-12 pre-adsorbed *S. aureus* ATCC 25923 to eliminate phage binding to the surface of G^+^ bacteria cells; then, they screened whole cells of *E. coli* ATCC 700928 to obtain a peptide with the sequence RLLFRKIRRLKR (EC5), which has an α-helix conformation. EC5 showed a strong antibacterial activity against G bacteria. The MICs against *E. coli*, *P. aeruginosa*, *Klebsiella pneumonia*, and other G^−^ bacteria ranged from 4 µg∙mL^−1^ to 128 µg∙mL^−1^. In addition, EC5 can destroy the plasma membrane of bacterial cells and thereby decrease the level of ATP and promote bacterial death.

#### 3.2.2. *Listeria Monocytogenes*

*L. monocytogenes* is a widely distributed pathogen that can cause arthritis and severe central nervous system infections, collectively known as listeriosis. Listeriosis is highly lethal and has been associated with foodborne epidemics over the past decade [64]. Z. Flachbartova et al. used the phage-display random-peptide library Ph.D.-12 to target MDR *L. monocytogenes* [23]. The selected peptides (L2 and L3) showed antibacterial activities against *L. monocytogenes*, and their MIC values were both 30 µM. They also reduced greatly the ATP level of *L. monocytogenes* at 1 × MIC. The exact antibacterial mechanisms of these peptides need to be further studied, and their listericidal activity, non-toxicity to mammalian cells, and lack of hemolytic activity suggest that they may have broad application prospects in the future.

#### 3.2.3. *S. aureus*

Passive immunization with antibodies generated against *S. aureus* may provide an effective means of preventing and treating *S. aureus* infection [65]. Man Wang et al. constructed a bovine scFv phage-display library with the cDNA of peripheral blood lymphocytes of dairy cows and *S. aureus*-induced mastitis. They used this library to screen the soluble whole-cell antigen of *S. aureus* (*S. aureus* cells were collected by centrifugation, washed, resuspended, sonicated, and centrifuged, and the supernatant was collected as soluble whole-cell antigen) [43]. Eight scFvs with high affinity to *S. aureus* antigen were obtained. All eight scFvs inhibited the growth of *S. aureus* in the culture medium, and at the inoculation amount of *S. aureus* was 10^8^ cfu∙mL^−1^, the growth inhibition effect of combinations of the eight scFvs was better than that of a single antibody. In addition, they confirmed the protective effect of these scFvs against *S. aureus*-induced mastitis in a mouse model. In brief, the authors reported a phage-display scFv library of bovine origin and discovered new bovine scFvs, which are expected to become novel therapeutic agents for the prevention of bovine mastitis caused by *S. aureus.*

Chicken egg yolk immunoglobulin antibodies are highly effective against *S. aureus* infectious diseases, but obtaining large amounts of highly purified specific antibodies is technically demanding and expensive [66]. To overcome all or several of these problems, Jingquan Li et al. used phage display to identify very specific and highly purified monoclonal fragments [41]. They immunized 100-day-old Roman chickens with inactivated *S. aureus*, amplified scFvs from the blood and spleen, and constructed T7 phage-display antibody library. *S. aureus* CVCC545 was used as the target, and four blood-derived scFvs and six splenic scFvs were obtained after three rounds of panning of T7 phage-display antibody library. The soluble protein SFV6 extracted from the spleen exhibited a good indirect ELISA response to *S. aureus* and can effectively inhibit its growth and proliferation.

Siji Nian et al. constructed a scFv library using mRNA from peripheral blood mononuclear cells of *S. aureus*-infected volunteers and screened the whole *S. aureus*; they obtained scFvs with a good binding activity toward *S. aureus*; then, they developed fully huscFv-Fcs that showed capability against *S. aureus*. These scFv-Fvs can be used as candidates for the development of future adjunctive therapy for severe *S. aureus* infections [40].

#### 3.2.4. *P. aeruginosa*

*P. aeruginosa* is an opportunistic human pathogen causing nosocomial and fatal infections in immunocompromised individuals. In 2017, *P. aeruginosa* was recognized as one of the most life-threatening bacteria, and it was listed as a priority pathogen needing new antibiotics by the World Health Organization. Hyun Kim et al. used the phage-display random-peptide library Ph.D.-12 to screen *P. aeruginosa* and identified a targeted peptide (PA2) that can specifically bind to OprF (a major OMP of *P. aeruginosa*) [15]. Although PA2 itself had no antibacterial activity, the hybrid peptide PA2-GNU7 constructed by coupling PA2 with GNU7 (a synthetic antimicrobial peptide [67]) can effectively inhibit the growth of *P. aeruginosa* at the MIC of 2 µM, which was a 16-fold increase in antimicrobial activity over that of GNU7. PA2-GNU7 was highly specific to and can kill *P. aeruginosa* rapidly and selectively in dual-species cultures of mid-logarithmic phase *P. aeruginosa* with either *E. coli* or *Salmonella typhimurium* cells at a 1:1 ratio. In a murine model of MDR *P. aeruginosa* infection, mice treated with 25 mg∙kg^−1^ PA2-GNU7 exhibited a survival rate of 100% after seven days, whereas those treated with 25 mg∙kg^−1^ meropenem exhibited a survival rate of 25%.

### 3.3. Identification of Molecular Interaction Sites

In addition to drug discovery, phage display technology is a powerful tool for the study of protein–protein, enzyme–substrate, and drug–protein interactions. Compared with traditional yeast two-hybrid technology, the phage display system has evident advantages in the study of molecular interactions, mainly including the following: (1) ability of phages to display great sequence diversity and (2) rapid panning in vitro, which shortens the research period. For the target protein of interest, researchers use a phage library to screen and identify its key functional domain or action site. This information is significant for drug design and modification and protein function research.

#### 3.3.1. R-TEM β-Lactamase

TEM-1 β-lactamases are the most common plasmid-encoded class A β-lactamases in G^−^ bacteria [68]. In class A β-lactamases, residues Ser70, Lys73, Ser130, and Glu166 play a major role in catalytic mechanism [69]. Identification of a large number of structural motifs that can maintain this catalytic activity is essential to the study of the evolutionary capability of enzymes and an important step toward the future design of clinically useful antibiotics and β-lactamase inhibitors [70]. To obtain an antibody library with the coverage of large antibody sequence diversity, Melody A. Shahsavarian et al. constructed a phage-display scFv library with a size of 2.7 × 10^9^ from classical murine strains BALB/C (healthy) and SJL/J (susceptible to developing autoimmune disease) [39]. The researchers used two different inhibitors of R-TEM β-lactamase as targets to screen catalytic antibodies with β-lactamase activity. Five antibody fragments were isolated, and all of them can hydrolyze the β-lactam ring. The structural simulation of the selected scFvs showed the presence of different motifs in each antibody fragment, which may be the reason for the differences in their catalytic activity. The result confirmed the plasticity of the active site of β-lactamases responsible for the wide resistance of these enzymes to clinically available inhibitors and antibiotics. These data provided valuable information on the potential structural possibilities capable of accommodating β-lactamase catalytic function and different antibiotic-resistance mechanisms mediated by various extended-spectrum β-lactamase (ESBL) enzymes, which have potential value for the further design of novel antibiotic candidates.

#### 3.3.2. β-Lactamase Inhibitory Protein (BLIP)

BLIP is a protein inhibitor of a diverse collection of class A β-lactamases produced by *Streptomyces clavulans*, which consists of 165 amino acids [71]. The interaction between BLIP and class A β-lactamases has become a model system for studying the amino acid sequence determinants of binding energy in a protein–protein complex because BLIP inhibits different class A β-lactamases with a wide range of affinities [72]. To identify mutants of BLIP with high binding affinity to PC1 β-lactamase and investigate the determinants of binding specificity, Ji Yuan et al. used purified *S. aureus* PC1 β-lactamase as a target to screen 23 BLIP combinatorial phage peptide libraries and found a BLIP K74G mutant that can inhibit PC1 β-lactamase [28]. The Ki (inhibition constant) of the BLIP K74G mutant was 42 nM, and the KD (dissociation constant) was 26 nM, which indicate that BLIP K74G mutant is a tight binding inhibitor of PC1 enzyme and can be used as a potential therapeutic or diagnostic reagent. Moreover, the study on the structure of BILP K74G–enzyme complex confirmed that the capability of salt bridge formation between position 74 of BLIP and position 104 of β-lactamase is an important determinant of binding specificity.

#### 3.3.3. Ribosome

Ribosomes, which translate genetic information into proteins, are critical organelles for the survival and growth of bacteria. They are composed of three rRNA chains (16S, 23S, and 5S) containing major functional sites and approximately 54 protein molecules. To date, about 60% of approved antibacterial agents work by targeting ribosomes [73]. Helix 31 (h31 or 970 loop) of bacterial 16S rRNA plays an important role in translation. The mutation study of h31 cyclic nucleotides showed that the ring structural changes caused by the disruption of critical stacking interactions strongly inhibited ribosome function in vivo [74]. Tek N. Lamichhane et al. obtained wild-type h31 binding peptides CVRPFAL and TLWDLIP using the phage-display random peptide library Ph.D.-7 for several rounds of screening [27]. CVRPFAL and TLWDLIP are desirable ligands for h31 with high binding affinities (KD values of 230 and 330 nM, respectively) and are equivalent to paramycin bound to the A site of 16S rRNA [75]; furthermore, CVRPFAL and TLWDLIP can inhibit protein synthesis of *E. coli* DH5 in vitro. These findings indicated the importance of h31 in protein synthesis and established h31 as a potential target for new drugs [74].

### 3.4. Detection or Diagnosis of Pathogenic Bacteria

Theoretically, any specific molecules or antigens on the surface of pathogens or whole bacterial cells can be screened to obtain a product that can specifically recognize pathogenic microorganisms and be used for early clinical diagnosis of pathogens.

Lipoarabinomannan (LAM) is a major component of the MTB cell wall and unique to *Mycobacterium* species, it has been considered as an ideal candidate for antigen-based tests. Masanori Kawasaki et al. used LAM as target to screen the scFv phage display library and discovered three mAbs (S4-20, G3, and TB) that can bind to epitopes unique in LAM from MTB and slow-growing nontuberculous mycobacteria (NTM) [33]. In 2021, Hong-Tao Zhang et al. developed novel rabbit anti-LAM IgG mAbs that can recognize LAM from slow-growing pathogenic mycobacteria. They constructed an immune scFv phage-display library using immunized rabbits with cell-wall components from the MTB H37Rv strain and screened the LAM to obtain a novel group of high-affinity rabbit anti-LAM mAbs. These mAbs had high sensitivities (100 pg∙mL^−1^) and affinities (1.16–1.73 × 10^−9^ M) toward LAM. They were also highly sensitive and reacted well with MTB H37Rv, *M. bovis*, slow-growing NTM species, and mycobacterial clinical isolates, which support their future clinical application [31].

Pep27 from *Streptococcus pneumoniae* is a small secretory, autolytic peptide containing 27 amino acid residues. The loss of Pep27 renders pneumococci avirulent and results in decreased levels of capsular polysaccharide, which establishes Pep27 as a pneumococcal virulence factor [76]. Pneumococcal surface protein A (PspA) is another pneumococcal virulence factor localized on the pneumococcal surface, and it can inhibit complement the activation and bactericidal activity of host lactoferrin [77]. Sangho Lee et al. used Pep27 and PspA as targets, and a synthetic huscFv library encoding HA-tagged scFv clones was used for biopanning. HuscFvs E2 and F9 can detect Pep27 in an environment mimicking in vivo conditions with KD values of 1.1 and 0.50 µM, respectively. HuscFv 2B11 can detect endogenous PspA from pneumococcal lysates with high affinity (KD = 5 nM). These HuscFvs can be used as molecular tool for biosensors to detect pneumococcal diseases [37,38].

Shilpakala Sainath Rao et al. used *B. cereus* 4342 cells as bait to screen a phage-display random-peptide library Ph.D.-12 and obtained two peptides: BBP-1 (AETVESCLAKSH) and BBP-2 (ALTLHPQPLDHP) [29]. They further confirmed that BBP-1 and BBP-2 bound specifically to *B. cereus* 4342 and *Bacillus anthracis* Sterne, respectively, because they did not cross-react with other *Bacillus* strains, thus, BBP-1 and BBP-2 can be used as models for *B. anthracis* detection under bio-safety level 2 conditions.

In addition, we can construct a special phage library to improve the specific recognition capability of the product to bacteria. The incorporation of a 2-acetylphenylboronic acid (APBA) warhead into cationic peptides can obtain selective probes of G^+^ bacteria, which readily label a target bacterium by way of a combination of reversible covalent and noncovalent interactions [78]. Kelly A. McCarthy et al. envisioned that introducing such reversible covalent warheads into phage libraries can establish a versatile platform to allow discovery of specific probes for diverse bacterial species and strains [21]. The researchers modified the phage display Ph.D.-C7C library and yielded the APBA-dimer library. A series of polypeptides was obtained by panning against the whole cell of *S. aureus* ATCC6538 with the APBA-dimer library, among which KAM5 showed significant binding capability toward *S. aureus*. The results of cell staining showed that KAM5 has a specific targeting binding effect on *S. aureus*. Researchers also panned the APBA-dimer library against a colistin-resistant strain of *Acinetobacter baumannii* and obtained polypeptide KAM8, which can target *A. baumannii* specifically. In 2020, several peptide probes that can detect colistin-resistant strains were obtained by utilizing this phage display platform [16]. These data indicated that this phage display platform enables rapid identification of peptide probes for a wide array of bacterial strains, contributing to the facile diagnosis and development of strain-specific antibiotics.

## 4. Conclusions and Future Perspective

In the past decade, phage display, as a quick method for obtaining candidate molecules with high affinity to specific targets, has become a powerful tool to develop drugs and diagnostic reagents for all kinds of diseases (e.g., cancer, diabetes, osteoporosis, multiple sclerosis, chronic pain, and infectious diseases). In addition, a significant research interest has focused on phage display in protein–protein interaction and protein function study, using it principally for identification of important target domains and binding sites. All these data have shown great values for the discovery of novel drug targets.

Compared with traditional drug discovery technology, phage display is a fast, simple, and low-cost identification method, which saves manpower and material resources. (1) Significantly improve the efficiency of finding candidate drug (peptides and antibodies): For example, more than 60% of the candidate drug molecules found by pharmaceutical chemical methods are not qualified in the process of “hit-to-lead“ for the reason that the affinity for the target is poor, while the phage display method has advantages in screening different targets to find drug candidate molecules with high potency [79]. (2) Efficiently screen ligands with high affinity for target with unknown functions: The target-based antimicrobial drug discovery strategies mostly rely on biochemical activity to screen inhibitors, so there is nothing to do when faced with proteins with unknown functions [80]. (3) Improve the efficiency of antibody drug research: Compared with the in vivo technology requiring animal immunity, the whole process of phage display technology is completely controllable, providing a high level of customization for direct and reasonable antibody discovery, thus significantly improving its therapeutic effect and reducing the use of animals.

However, several problems have been encountered in the development and application of phage display technology in the antibacterial field.

First, obtaining agents with very strong bactericidal activity against host bacteria (usually *E. coli*) is difficult. During the entire panning process, the proliferation of phage and the expression of surface proteins or polypeptides depend on host bacteria (usually *E. coli*). Several reports showed peptides with inhibitory effect on *E. coli*, but their MIC or MBC were often extremely high; therefore, if we are to use this technology to screen powerful bactericides, target selection is very important. We can eliminate potential targets that are highly expressed or vital to the survival of the host bacteria by comparison with the gene information of the host bacteria; thus, virulence factors or drug resistance proteins may be suitable targets for phage display screening.

Second, M13, the most widely used phage display system in the field of antibacterials, still has shortcomings, with the most important one being the translocation of fusion proteins in an unfolded state to the periplasm in M13 phages. In addition, fusion and incorporation of the phage coat protein with its fusion partner occur in the periplasm. This feature is an advantage for several proteins, including secreted heterologous proteins but a disadvantage for cytoplasmic proteins or molecules that have an inhibitory effect on the secretion process, which cannot pass through the inner membrane and cannot be transported to the periplasm to complete the folding and be finally displayed correctly on the phage surface [81]. To overcome the deficiency of M13, scientists have focused on virulent phage T7 in the field of pathogenic microorganisms. On the one hand, the display of fusion proteins in T7 was independent of its capability to transport across the cell membrane. On the other hand, the T7 phage display system remained extremely robust and stable under conditions where other phages may be inactivated [82,83]. Thus, the T7 display system is likely to play a great role in the field of antimicrobial research in the future.

Furthermore, although phage display technology is very effective in discovering mAbs and peptides, the corresponding screening procedures are often challenged by false positive clones, which are often found in the selection step. These false positive clones remain a major obstacle in library screening because the reason for their enrichment is difficult to define and deal with [84].

In addition, due to its enzymatic instability, unmodified peptides can be rapidly cleared from plasma within minutes if they enter the human body, making it difficult for them to play long-term pharmacological effects, and their drug-forming properties are low, which leads to the very limited use of peptides in clinical practice. At present, for this problem, cyclic peptide libraries and bicyclic peptide libraries have been developed to increase the stability of the peptides obtained by screening.

Despite the above defects of phage display technology, we believe that with its continuous improvement and development, it will play a more active role in the treatment of bacterial infectious diseases.

## Figures and Tables

**Figure 1 molecules-28-02621-f001:**
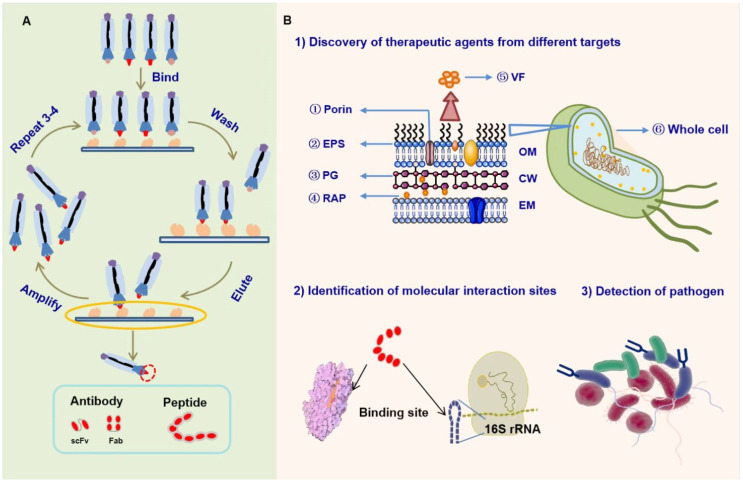
The overview of phage display technology. (**A**) The basic process of phage display screening experiment. (**B**) Application of phage display technology in the antibacterial field [6]. (1) Discovery of therapeutic peptides or antibodies from a single molecular target (e.g., ETA, Penicillinase, LpxA, and so on) or the whole cell; (2) identification of molecular interaction sites; and (3) detection or diagnosis of pathogen. VF: virulence factors; PG: peptidoglycan; OM: outer membrane; CW: cell wall; and IM: inner membrane.

**Table 1 molecules-28-02621-t001:** Application of peptide phage display technology in antibacterial field.

Target	Bacterial Strain	Library	Product	Potential Application	Time	Reference
OprF	*P. aeruginosa*	Ph.D.-12 peptide library	PA2 (SQRKLAAKLTSK) PA2-GNU7 (SQRKLAAKLTSK-GGGRLLRPLLQLLKQKLR)	Antimicrobial candidates	2020	[15]
CR bacteria	CR bacteria	APBA-dimer peptide library	MAK30 (ACmDPNRMDRCm) SEC5(ACmKPLHSRSCm) SEC18(ACmSERQHLQCm)	Detection of bacteria	2020	[16]
Cep-1 Ala-2 Lac-4	*S. aureus* MRSA VRE	Bicyclic peptide libraries (BC-A, BC-B)	P14 (CQTDVCQRTIC) P15 (CSLITQCGGVGC) P18 (CGGGICRTHNC)	Antimicrobial candidates	2019	[17]
LpxA	*E. coli*	Recombinant M13 phage library	P920 (SSGWMLDPIAGKWSR) CR20 (WMLDPIAGKWSR)	Antimicrobial candidates	2019 2003 2006	[18,19,20]
*S. aureus*	*S. aureus*	APBA-dimer peptide library	KAM5 (ACmVSPRSHECm)	Detection of *S. aureus*	2018	[21]
*A.baumannii*	*LOS−A. baumannii*	APBA-dimer peptide library	KAM8 (ACmTLPNGPRCm)	Detection of *LOS− A. baumannii*	2018	[21]
Lipid II	G^+^ bacteria	Bicyclic peptide library	P8 (ACLLQSLLCPYSTHRCG) P8-D-C10 lipopeptides	Antimicrobial candidates	2017	[22]
MDR *L. monocytogenes*	MDR *L. monocytogenes*	Ph.D.-12 peptide library	L2 (DQFVHDVKGTKH) L3 (NSWIQAPDTKSI)	Antimicrobial candidates	2016	[23]
Penicillinase	ND	Ph.D.-12 peptide library	P2 (NIYTTPWGSNWS)	Antimicrobial candidates	2015	[24]
*E. coli*	*E. coli*, *P. aeruginosa*	Ph.D.-12 peptide library	EC5 (RLLFRKIRRLKR)	Antimicrobial candidates	2013	[25]
R-HPr protein	*E. coli* DH5α	Ph.D.-12 peptide library	AP1 (YQVTQSKVMSHR)	Antimicrobial candidates	2012	[26]
h31	*E. coli* DH5	Ph.D.-7 - peptide library	CVRPFAL, TLWDLIP	Antimicrobial candidates	2011	[27]
PC1 β- lactamase	*S. aureus*	BLIP combinatorial peptide libraries	BLIP K74G mutant	Further research for BLIP residues	2011	[28]
*B. cereus*	*B. anthracis*	Ph.D.-12 peptide library	BBP-1 (AETVESCLAKSH) BBP-2 (ALTLHPQPLDHP)	Detection of *B. anthracis*	2010	[29]

Abbreviations: CR, Colistin-resistant; Cm, APBA-IA modified cysteine; *LOS− A. baumannii*, lipooligosaccharide deficient *A. baumannii*; ND, Not detect; R-HPr, Recombinant HPr.

**Table 2 molecules-28-02621-t002:** Application of antibody phage display technology in antibacterial field.

Target	Bacterial Strain	Library	Product	Potential Application	Time	Reference
ICL	MTB	HuscFv library	a-rICL-C6	Antimicrobial candidates	2022	[30]
LAM	MTB	Immunized rabbit scFv library	Rabbit anti-LAM IgG mAbs	Detection of MTB	2021	[31]
OmpU	*V.* *parahaemolyticus*	Human sdAb library	UAb28	Antimicrobial candidates	2020	[32]
LAM	MTB	Immunized rabbit and chicken scFv library	Three mAbs (S4-20, G3, and TB)	Detection of MTB	2019	[33]
SpuE	*P. aeruginosa*	HuscFv library	ScFv5	Antimicrobial candidates	2019	[34]
ETA	*P. aeruginosa*	HuscFv library	ETA bound-HuscFvs	Antimicrobial candidates	2019	[35]
Flap region of urease enzyme	*H. pylori*	Human semi-synthetic antibody library	V_L_ antibody C3	Antimicrobial candidates	2019	[36]
Pep27	*S. pneumoniae*	HuscFv library	Two HuscFvs (E2 and F9)	Detection of *S. pneumoniae*	2018	[37]
PspA	*S. pneumoniae*	HuscFv library	HuscFv 2B11	Detection of *S. pneumoniae*	2017	[38]
Pep90, penam sulfone derivative	ND	Immunized mice scFv library	P90C1, P90C2, P90C3, PSC1 and PSC2	Further information on the structure about β-lactamase catalytic function	2017	[39]
*S. aureus*	*S. aureus*	HuscFv library	ScFv-Fcs (S78, S117)	Antimicrobial candidates	2016	[40]
*S. aureus*	*S. aureu*	Immunized chicken T7 scFv antibody library	SFV6	Antimicrobial candidates	2016	[41]
MrkA	*K. pneumoniae*	HuscFv library	IgG1 (KP3, KP16)	Antimicrobial candidates	2016	[42]
*S. aureus* soluble whole-cell antigen	*S. aureus*	Immunized cows bovine scFv library	8 anti- *S. aureus* scFvs	Antimicrobial candidates	2016	[43]
VIM-4 MβL	ND	Immunized llama and dromedary nanobody library	NbVIM_38 nanobody	Antimicrobial candidates	2013	[14]

Abbreviations: ND, Not detect.

## Data Availability

Not applicable.

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
