# Peer review of "Phage Display-Derived Peptides and Antibodies for Bacterial Infectious Diseases Therapy and Diagnosis"

_molecules, 2023, doi:10.3390/molecules28062621_

Round 1
Reviewer 1 Report
The review by Zhao and colleagues summarized recent progress in phage display for the discovery of novel therapeutic agents, identification of action sites of target proteins of bacteria, and rapid detection of pathogens, but also discussed several shortcomings of this technology that need to be resolved. Phage display technology, with its continuous improvement and development will play an increasingly significant role in the treatment of bacterial infections. The manuscript could be published after addressing the following issues.
1) Tables 1 and 2 should include references.
2) In figure 1, references should be added.
3) The references should be double checked to ensure that they follow the journal's style.
4) English should be improved.
Author Response
Thank you for your positive comments on our manuscript, we had revised the manuscript according to your helpful suggestions.
1) We had added the references to the table 1 and table 2.
2) We had added the references to the figure 1.
3) We had revised the reference format according to the requirements of the journal.
4) We had polished the grammar carefully and worked on readability of our manuscript.
We provided a file named molecules-2247630_revised.docx that marked in red where we revised.
Reviewer 2 Report
This review provides a systematic and up-to-date overview on the Phage display-derived peptides and antimicrobial applications for different strategies. The selected topic is of great significance and of high interest to a broad readership in the fields of antibacterial, peptide drugs, synthetic biology, etc. I recommend this review paper for publication in Molecules after a minor revision. Detailed comments are as follows:
1. The advantages of phage display technology over traditional approach for antimicrobial drug or functional peptide discovery should be summarized.
2. In application of phage display in the antibacterial field,the author summarized many presently reported strategies. It is better to comment and compare these different strategies so that readers can gain a more systematic understanding of this field.
3. M. tuberculosis is mentioned several times in Table 2, relevant contents should be introduced in 3.2.
4. More figures of some classic works or important developments should be included.
5. More literatures should be cited in the “Introduction” and the “Overview of Phage Display Technology” parts.
6. Some other literatures about the application of antibacterial peptides should be cited. Such as, Peptide hydrogel based sponge patch for wound infection treatment, Front. Bioeng. Biotechnol., 2022 Dec 15;10:1066306. Near-Infrared Light Brightens Bacterial Disinfection: Recent Progress and Perspectives, ACS Appl. Bio Mater. 2021, 4, 3937.
Author Response
- The advantages of phage display technology over traditional approach for antimicrobial drug or functional peptide discovery should be summarized.
Response: Thank you for your suggestion. We have added the advantages of phage display technology in the discussion part of our manuscript :
“Compared with traditional drug discovery technology, phage display is a fast, simple and low-cost identification method, which saves manpower and material resources. 1) Significantly improving the efficiency of finding candidate drug (peptides and antibodies): For example, more than 60% of the candidate drug molecules found by pharmaceutical chemical methods are not qualified in the process of " hit-to-lead ", for the reason that the affinity for the target is poor, while the phage display method has advantages in screening different targets to find drug candidate molecules with high potency. 2) Efficiently screening ligands with high affinity for target with unknown functions: The target-based antimicrobial drug discovery strategies mostly rely on biochemical activity to screen inhibitors, so there is nothing to do when faced with proteins with unknown functions. 3) Greatly promoting the research of antibody drugs: Compared with the in vivo technology requiring animal immunity, the whole process of phage display technology is completely controllable, providing a high level of customization for direct and reasonable antibody discovery, thus significantly improving its therapeutic effect and reducing the use of animals.”
- In application of phage display in the antibacterial field,the author summarized many presently reported strategies. It is better to comment and compare these different strategies so that readers can gain a more systematic understanding of this field.
Response: Thank you for your suggestion. In this paper, we compare the strategies of phage display technology commonly used in the antibacterial field, and extract relevant contents as follows:
First, we compared the whole cell and non-whole-cell based screening strategies (In the second paragraph of Part 3. Application of Phage Display in the Antibacterial Field): “In comparison with specific molecular targets, whole-cell-based screening may be more complex. However, this type of screening has a potential advantage to recognize cell surface structures that have not been previously identified, and all targets on the cell surface are screened simultaneously in their native physiological context. ”
Secondly, we compared the screening strategies targeting different molecules:
1)Targeting of bacterial virulence factors (In the first paragraph of Part 3.1.1. Targeting of bacterial virulence factors): “Unlike traditional bacteriostatic or bactericides, anti-virulence agents generally target the key molecules involved in bacterial pathogenic mechanisms and decrease the bacterial pathogenicity, allowing bacteria to be cleared easily by the host immune system. Therefore, anti-virulence molecules have minimal significant influence on the survival/adaptability of bacteria and reduce the pressure of inducing bacterial resistance. In addition, they cause no damage nor alteration in the composition of natural host microbiota, which implies improvement in long-term safety. Thus far, a large number of experimental data have confirmed that anti-virulence strategies can be successfully used to combat bacterial infections.”
2)Targeting resistance associated proteins (In the first paragraph of Part 3.1.2. Targeting resistance associated proteins): “Through inhibition of drug-resistant proteins or enzymes, targeting resistance associated proteins is an effective way to overcome acquired resistance mechanisms of bacteria and prolong the lifespan of existing antibiotics. These anti-resistance agents, such as clavulanic acid and tazobactam, lack or have very weak antibacterial activity and generally cannot be used alone. However, not all resistance-related molecules are suitable targets for antimicrobial drug development because bacteria are extremely cunning and develop diverse strategies to adapt to environmental changes. Therefore, numerous bacteria exhibit multiple drug resistance mechanisms, and obtaining the desired effect through intervention of a single mechanism may be difficult.”
3)Targeting proteins necessary for bacterial survival (In the first paragraph of Part 3.1.3. Targeting proteins necessary for bacterial survival): “Different from anti-virulence and anti-resistance strategy, targeting key molecules necessary for bacterial survival has been the mainstream direction of antimicrobial research because this strategy can directly inhibit or kill bacteria and has significant therapeutic value in clinical practice. However, based on traditional antibacterial targets, such as penicillin-binding proteins, finding new compounds that are better than existing antibiotics has been difficult. Therefore, developing antibacterial drugs with new antibacterial mechanisms, that is, by acting on new targets, are one of the most important strategies for controlling MDR bacteria.”
- M. tuberculosis is mentioned several times in Table 2, relevant contents should be introduced in 3.2.
Response: Thank you for your suggestion. We have revised this part of our manuscript as follows (In the section4 of Part 3.1.3. Targeting proteins necessary for bacterial survival):
“Despite the existence of effective combination chemotherapy and neonatal anti-tuberculosis (TB) vaccine, the clinical diagnosis and treatment of TB caused by Mycobacterium tuberculosis (MTB) infection are still complex and difficult, which is one of the serious challenges facing global public health.”
- More figures of some classic works or important developments should be included.
Response: Thank you for your suggestion. This manuscript mainly reviewed the application progress of phage display technology in the discovery of antibacterial molecules, therefore, on the one hand, we think that the detailed data and information are better displayed in tables, and only one figure is displayed to introduce the screening process of phage display technology and its main application in the field of antibacterial field; On the other hand, we believe that the construction method of phage display library or the specific mechanism of action of a molecule and target are not the focus of this paper, so the relevant figures are not quoted, and only the references are listed in the necessary position.
- More literatures should be cited in the “Introduction” and the “Overview of Phage Display Technology” parts.
Response: Thank you for your suggestion. We had added several references in the “Introduction” and the “Overview of Phage Display Technology” parts, such as reference [1], reference [6], reference [9], reference [10], reference [11].
- Some other literatures about the application of antibacterial peptides should be cited. Such as, Peptide hydrogel based sponge patch for wound infection treatment, Front. Bioeng. Biotechnol., 2022 Dec 15;10:1066306. Near-Infrared Light Brightens Bacterial Disinfection: Recent Progress and Perspectives, ACS Appl. Bio Mater. 2021, 4, 3937.
Answer:Thank you for your suggestion. We have downloaded and carefully read the two articles you recommended, and found that:
(1) In the first paper (Front. Bioeng. Biotechnol., 2022 Dec 15;10:1066306.), a new type of dressing based on polypeptide functional sponge patch was constructed. The porous sponge patch is made of antimicrobial peptide and medical agarose through gel and freeze-drying technology. The results showed that the porous sponge has excellent antibacterial and anti-skin infection activities. But the peptide used in this paper (AKF-12H) was not discovered by phage display technology, so it cannot be cited in our manuscript.
(2) In the second paper (ACS Appl. Bio Mater. 2021, 4, 3937.), the author reviewed the recently developed near-infrared (NIR) light-irradiation-based bacterial disinfection, which is highly promising to shatter bacterial resistance by employing NIR light-responsive materials as mediums to generate antibacterial factors such as heat, reactive oxygen species (ROS), and so on. However, this treatment approach is not related to the topic of our manuscript. We focused the recent progress of phage display for the discovery of novel therapeutic agents, identification of action sites of bacterial target proteins, and rapid detection of different pathogens. So it was not cited in our manuscript.
Reviewer 3 Report
The paper is devoted to the actual problem of finding antibacterial drugs against multidrug-resistant and super-bacteria. Phage display represent one of the most powerful methods for the development of antibacterial molecules based on their binding to bacterial targets. A particular advantage of phage displays is that the amino acid sequences found can be varied and supplemented with others to increase their effectiveness.
The strengths of the review are the systematization of the method in conjunction with different target classes, as well as the discussion of problems of this technology.
The reviewer has a few minor remarks:
1. Section 2 contains the phrase "Antibody phage display allows the immediate generation of full human antibodies". This phrase is somewhat erratic, since in order to generate full human antibodies based on the screening of single chains from phase displays, non-trivial steps are needed to convert them into full-length Ig.
2. Figure 1 schematically shows the library elements that are displayed on phages. In this case, an antibody composed of 4 chains is drawn. It would be better to schematically draw a single chain protein as scFvs or Fabs, but not as a classical antibody.
The reviewer recommends publishing this review.
Author Response
1. Section 2 contains the phrase "Antibody phage display allows the immediate generation of full human antibodies". This phrase is somewhat erratic, since in order to generate full human antibodies based on the screening of single chains from phase displays, non-trivial steps are needed to convert them into full-length Ig.
Response: Thank you very much for your constructive comments concerning our manuscript, according to your suggestion, we have revised this sentence as follows (In the third paragraph of Part 2 Overview of Phage Display Technology ):
” Antibody phage display allows researchers to obtain human antibodies more quickly, because theoretically, antibodies can be found through in vitro screening, without relying on immunity and related restrictions.”
2. Figure 1 schematically shows the library elements that are displayed on phages. In this case, an antibody composed of 4 chains is drawn. It would be better to schematically draw a single chain protein as scFvs or Fabs, but not as a classical antibody.
Response: Thank you for your suggestion. We have replaced the classical antibody with scFvs or Fabs in figure 1.
We provided a file named Manuscript_revised.docx that marked in red where we revised